# Pyruvate Kinase Deficiency: Markedly Decreased Reticulocyte PK Activity and Limited Specificity of the PK/HK Ratio

**DOI:** 10.3390/ijms26178606

**Published:** 2025-09-04

**Authors:** Larisa Koleva, Ivan A. Dolgikh, Aleksandra V. Kryukova, Dmitry S. Prudinnik, Elizaveta A. Bovt, Soslan S. Shakhidzhanov, Svetlana G. Mann, Nataliya S. Smetanina, Fazoil I. Ataullakhanov, Elena I. Sinauridze

**Affiliations:** 1Dmitry Rogachev National Medical Research Center of Pediatric Hematology, Oncology and Immunology, 1 Samor Mashel Str., 117198 Moscow, Russia; ivan.dolgih@dgoi.ru (I.A.D.); maddima285@gmail.com (D.S.P.); elizaveta.bovt@dgoi.ru (E.A.B.); soslan.shakhidzhanov@dgoi.ru (S.S.S.); svetlana.mann@dgoi.ru (S.G.M.); nataliya.smetanina@dgoi.ru (N.S.S.); sinauridze.elena@dgoi.ru (E.I.S.); 2Center for the Theoretical Problems of Physico-Chemical Pharmacology, Russian Academy of Sciences, 30 Srednyaya Kalitnikovskaya Str., 109029 Moscow, Russia; ataullakhanov.fazly@gmail.com; 3Faculty of Fundamental Physical and Chemical Engineering, M.V. Lomonosov Moscow State University, 1 Leninskie Gory Str., 119991 Moscow, Russia; shkryukova@gmail.com; 4Department of Physiology, Perelman School of Medicine, University of Pennsylvania, 3400 Civic Center Blvd, Philadelphia, PA 19104, USA

**Keywords:** red blood cells, pyruvate kinase deficiency, PK activity in reticulocytes, PK/HK ratio, differential diagnosis of PK deficiency, hereditary hemolytic anemias

## Abstract

Diagnosis of pyruvate kinase deficiency (PKD) remains challenging in clinical practice. The pyruvate kinase (PK) to hexokinase (HK) activity ratio (PK/HK) was proposed to reduce the confounding effect of reticulocytosis on PK activity measurement. However, decreased PK activity and PK/HK ratios have also been observed in other anemias, raising doubts about their diagnostic value. We assessed the diagnostic accuracy of PK/HK ratio versus PK activity in differentiating PKD from other hereditary anemias. This study included 41 patients with molecularly confirmed PKD and 62 patients with other anemias. We also evaluated the influence of reticulocytosis and transfusions on erythrocyte PK activity. The PK/HK ratio showed 73% specificity, while PK activity alone achieved 95%. In PKD patients, reticulocytosis did not affect PK activity because reticulocyte PK activity was already markedly reduced (23-fold) compared with controls. In other anemias, decreases in PK activity were present in both reticulocytes and erythrocytes, but to a lesser extent. Transfusions contribute more to the false-normal result of PK activity than reticulocytosis. Measuring reticulocyte-specific PK activity during regular transfusions provided reliable results, as only patient-derived reticulocytes are present in the blood. PK activity demonstrates higher specificity than PK/HK ratio in diagnosing PKD. Reticulocytosis is not a confounder, while transfusions remain the main limitation. Reticulocyte-specific PK activity measurement may improve diagnostic accuracy in transfused patients.

## 1. Introduction

Pyruvate kinase (PK) deficiency is the most common hereditary non-spherocytic hemolytic anemia caused by a defect in erythrocyte glycolytic enzymes. It is considered an underdiagnosed condition with a reported prevalence of approximately 51 per million population [1].

Pyruvate kinase deficiency (PKD) leads to the accumulation of upstream glycolytic intermediates [2], increased erythrocyte volume and, consequently, premature hemolysis and anemia [3]. The disorder is caused by mutations in the *PKLR* gene on chromosome 1q21 [4]. Clinical symptoms are restricted to individuals with biallelic pathogenic variants (homozygous or compound heterozygous) and show a broad phenotypic spectrum [5]. Both the clinical presentation of PKD and routine hematologic parameters are nonspecific. They are typical for chronic hemolysis seen in other forms of hereditary hemolytic anemias such as hereditary spherocytosis, stomatocytosis, autoimmune hemolytic anemia, etc. [6,7]. As a consequence, this complicates the differential diagnosis of PKD [8]. The diagnosis ultimately depends upon the demonstration of decreased enzyme activity and the identification of causative mutations in the *PKLR* gene [7]. However, up to 20% of patients carry novel variants of uncertain significance, and in approximately 10% of cases, no mutations are identified through routine exon sequencing [6,9,10]. Additionally, genetic analysis remains costly and not always readily accessible.

The enzymatic diagnosis of PKD is complicated by a number of preanalytical and biological factors, including chronic transfusion therapy, reticulocytosis, and leukocyte contamination. It is generally believed that all three factors may cause overestimation of measured PK activity and lead to a false normal result [7,11]. The impact of leukocyte contamination has been comprehensively discussed by Bianchi et al. [7] and is not addressed in this study. Reticulocytosis can result in overestimation as the activity of several glycolytic enzymes including PK is markedly higher in reticulocytes compared to mature erythrocytes [12,13,14]. Although A. Zanella et al. [15] reported no direct correlation between PK activity and reticulocyte count in PKD patients, reticulocytosis remains a diagnostic concern. Consequently, current diagnostic guidelines increasingly recommend the use of the PK to hexokinase (HK) activity ratio (PK/HK), since HK also demonstrates higher activity in reticulocytes [6,9,16,17]. A decreased PK/HK ratio is typically interpreted as suggestive of pyruvate kinase deficiency in the presence of reticulocytosis. Several studies have reported higher diagnostic sensitivity of the PK/HK ratio compared to PK activity alone [9,16,17]. However, these assessments were limited to comparisons between PKD patients and healthy controls. To validate the reliability of PK/HK ratio as a tool for differential diagnosis of PKD, it is necessary to evaluate the sensitivity and specificity of this parameter by including patients with other anemias. Decreased PK activity and reduced PK/HK ratios have also been reported in other disorders, including hereditary spherocytosis [18,19], xerocytosis and beta-thalassemia [20], sickle cell anemia [20,21,22], myelodysplastic syndrome [23], and Diamond–Blackfan anemia [24]. In these conditions, which are also commonly associated with reticulocytosis, decreased PK activity may lead to a falsely low PK/HK ratio, thereby complicating the differential diagnosis between PKD and other anemias with overlapping clinical features.

There is currently a lack of published data on the diagnostic sensitivity and specificity of the PK/HK ratio for differentiating PKD from other anemias that commonly undergo PK activity testing due to similar clinical features. Moreover, several relevant questions remain open:The actual level of PK activity in the reticulocytes of PKD patients and the impact of reticulocytosis on the total PK activity (activity in erythrocytes isolated from whole blood purified from leukocytes);The diagnostic approach for PKD patients undergoing regular monthly transfusions, who comprise between 11% and 53% of the PKD population depending on age [4,25];The reliability of PK activity and the PK/HK ratio for differential diagnosis between PKD and other anemias with similar clinical presentations and reduced PK activity;The critical level of residual PK activity associated with clinically relevant hemolysis and anemia.

We evaluated the diagnostic sensitivity and specificity of PK activity and the PK/HK ratio in a large cohort of patients, in comparison not only with healthy donors but also with patients suffering from other anemias. This approach is critical for the differential diagnosis of anemias, as many patients without PKD are often referred for PK testing before genetic results are available. Moreover, we assessed the contribution of reticulocytosis and transfusions to measured PK activity. Then we determined the specific PK activity in both reticulocytes (A_R_) and mature erythrocytes (A_E_) in patients with PKD and other anemias and demonstrated that PK activity is markedly reduced in reticulocytes of PKD patients. Based on these findings, we propose a practical diagnostic approach for identifying PKD in regularly transfused patients.

## 2. Results

This study included 95 healthy donors and 108 patients, of whom 46 had PKD (13 were homozygous, 33 were compound heterozygous, including the siblings (No. 10 and 11; 20 and 21; 32 and 33; 39 and 40 (Table 1)). The remaining 62 patients had other types of anemia, of which 44 patients (71%) had various hereditary erythrocyte membranopathies, 5 patients had other enzymopathies, and 7 patients had hemoglobinopathies, as well as cases of sideroblastic and megaloblastic anemia, paroxysmal nocturnal hemoglobinuria, Diamond–Blackfan anemia, congenital dyserythropoietic anemia, atypical hemolytic uremic syndrome, and myelodysplastic syndrome. Appendix A presents the clinical, hematological, and molecular data for all patients. The median age of the patients was 10.5 years (range 1–61), and 15% of the patients were splenectomized.

### 2.1. Sensitivity and Specificity of the PK Activity Assay for the Differential Diagnosis of PK Deficiency

PK activity was measured in all patients included in this study as well as in healthy controls (Figure 1A). The reference interval was 8.4–16.4 IU/gHb, with a median of 10.85 IU/gHb. In 42/46 (91%) patients with PKD and in 7/62 (11%) patients with other anemias, PK activity was below or at the lower limit of the normal range. The median PK activity was 3.9 IU/gHb (range 0.83–12.14) and 13.49 IU/gHb (range 5.66–27.65) for patients with PKD and those with other anemias, respectively. The diagnostic sensitivity and specificity of PK activity were 91% and 95%, respectively, at a cut-off value of 8.44 IU/gHb (AUC 0.98, LCL 0.959, UCL 0.995).

Recent (≤3 months) erythrocyte transfusions may contribute to the diagnostic sensitivity and specificity of PK activity. If the PKD patients with recently received transfusions are excluded from the calculations, the diagnostic sensitivity of PK activity increases to 97% (AUC 0.99, LCL 0.987, UCL 1.003). In this case, only 1/29 (3.7%) PKD patients had PK activity within the normal range (Figure 1B).

The reference range for residual PK activity is 78–151% (Figure 1B). The median residual PK activity among patients with other anemias was 124%. Residual PK activity below the normal range or at its lower limit was observed in 11% of patients with anemias such as HK deficiency combined with β-thalassemia (n = 1, No. 60 in Appendix A), hereditary spherocytosis (n = 3, No. 102, 104, 107), unspecified membranopathy (n = 1, No. 50), hereditary stomatocytosis (n = 1, No. 62), and sideroblastic anemia (n = 1, No. 83).

In some patients with other anemias, PK activity can be decreased due to secondary effects of the underlying disease. This raises the question of how severe the decrease in PK activity must be to be considered a primary cause of anemia (i.e., PKD). In 28/29 (97%) patients with PKD without recent transfusions, residual PK activity did not exceed 66%, with a median of 27%. In 3/62 (5%) patients with other anemias residual PK activity was also below 66%. Only one patient with PKD (No. 14) had a residual PK activity of 81%. This patient carries a variant of uncertain clinical significance (c.1130T>C) in one allele. Notably, residual PK activity in patients with PKD carrying known pathogenic mutations (n = 24) did not exceed 54%, with a median of 18%. Residual PK activity above 54% was observed in 5/29 patients with PKD (Table 1, No. 14, 36, 39, 40, 42) without recent transfusions, each carrying at least one variant of uncertain significance (Figure 1B, yellow dots).

Thus, in 97% of cases without recent transfusions, residual PK activity above 66% indicates an anemia not associated with PKD, even if PK activity is below the normal range. At a cut-off of 67%, the diagnostic sensitivity and specificity of residual PK activity were 97% and 98%, respectively. When excluding patients with variants of uncertain clinical significance from the calculations, the diagnostic sensitivity and specificity of residual PK activity were 100% and 99.4%, respectively, at a cut-off of 56.9%.

### 2.2. Comparison of Sensitivity and Specificity of PK Activity and the PK/HK Ratio in the Diagnosis of PKD

For 25 patients with PKD, 15 patients with other anemias, and 14 healthy donors, hexokinase (HK) activity in erythrocytes was also measured (Figure 2A) and the PK/HK ratio was calculated (Figure 2B). For these patient groups, PK activity below the normal range (8.4 IU/gHb) was observed in 24/25 patients with PKD (1 patient had normal PK activity due to a recent erythrocyte transfusion). In 3/15 patients with other anemias, PK activity was below or at the lower limit of normal value.

The reference interval for the PK/HK ratio was 9.9–15.7. A PK/HK ratio below the normal range was observed in 23/25 (92%) patients with PKD (median of 2.75) and in 7/15 (47%) patients with other anemias (median of 10.84). At a cut-off value of 10 (the lower limit of the normal range), the PK/HK ratio showed 92% sensitivity and 73% specificity (AUC 0.96, LCL 0.918, UCL 1.008), making it less specific than PK activity (91% sensitivity, 95% specificity) for diagnosing PKD (difference between AUCs was 0.0124 ± 0.0154).

HK activity decreases with the age of erythrocytes [12] and we observe a direct correlation with reticulocyte count in blood in both PKD and other anemias (Figure 3). Therefore, even with slightly reduced PK activity and reticulocytosis, the PK/HK ratio may be decreased in patients with other anemias. Nevertheless, the PK/HK ratio for patients with PKD without recent transfusions is lower than for most patients with other anemias. Thus, the PK/HK ratio can be used in the diagnosis of PKD, but the reference range should be determined considering the PK/HK ratio in patients with other anemias.

For our patient group, the specificity of the PK/HK ratio for the differential diagnosis of PKD increases to 97% while maintaining the same sensitivity when selecting a cut-off of 7.67 for the ratio, which is below the normal values for healthy donors (10). Additionally, in the presence of red blood cell transfusions, the values of the PK/HK ratio, as well as PK activity in erythrocytes, can be falsely normal.

### 2.3. The Relationship Between Measured PK Activity and Reticulocyte Count in the Blood

To assess the contribution of reticulocytosis to the diagnosis of PKD, we examined the relationship between PK activity and reticulocyte count in the blood for 41 patients with PKD, for whom reticulocyte percentages were available on the date of PK activity analysis (Figure 4A). No correlation was observed between PK activity and reticulocyte levels in either transfused or non-transfused PKD patients.

Regardless of the reticulocyte percentage, PK activity remained below the normal range in all patients with PKD, except for those who had recently received red blood cell transfusions (blue dots in Figure 4A), the majority of whom had reticulocyte counts not exceeding 5%.

In contrast, when examining patients with other anemias (Figure 4B), a positive correlation (r = 0.42, at *p* = 0.0013) between PK activity and reticulocyte count was observed. This finding is consistent with previous reports indicating higher PK activity in reticulocytes of healthy controls [12].

### 2.4. Specific PK Activities in the Reticulocyte and Erythrocyte and Their Ratio

For 24/46 patients with PKD, 25/62 patients with other anemias, and 10 healthy controls, PK activity was measured in several erythrocyte fractions with different reticulocyte contents, obtained using a Percoll density gradient. Based on the graphical relationship between PK activity and reticulocyte count in fractions of red blood cells with different reticulocyte content (see Section 4), specific PK activities in reticulocytes (A_R_) and erythrocytes (A_E_) were determined for each patient and donor (Figure 5A–C, Table 1). For patients with recent transfusions, A_E_ data were excluded from the graphs and calculations of the A_R_/A_E_ ratio.

It was found that in patients with PKD, the specific PK activity in reticulocytes (median 10.3 IU/gHb) was reduced 23-fold compared to that of healthy controls (median 231.5 IU/gHb) (Figure 5A,B). Interestingly, in patients with anemias not associated with PKD, specific PK activity in reticulocytes was also reduced (median 88.3 IU/gHb), but to a lesser extent (2.6-fold) (Figure 5C).

The ratio of specific PK activity in reticulocytes to erythrocytes (A_R_/A_E_) varied across the different groups (Figure 5D). Healthy donors showed a more pronounced difference between activity in reticulocytes and erythrocytes compared to patients. The median A_R_/A_E_ ratio for patients with PKD was 4.5 (range 1.97–14.93), while for patients with other anemias, it was 12.7 (range 1.94–34.4), and for healthy donors it was 26.7 (range 7.3–65). For 70% of patients with PKD, the A_R_/A_E_ ratio was ≤5.4 (Figure 5D, Table 1).

Table 1 presents the data on total and specific PK activity in reticulocytes and erythrocytes, as well as the PK/HK ratio for patients with PKD and other anemias.

### 2.5. The Impact of Donor Red Blood Cells Transfusions on the PKD Diagnosis

As shown above (Figure 4A), reticulocytosis has limited influence on the total PK activity in patients with PKD, whereas recent transfusions significantly increase this parameter (Figure 6). Figure 6 illustrates PK activity in PKD patients with and without recent (≤3 months prior to assay) red blood cell transfusions.

It is clearly seen that the median PK activity in patients without donor red blood cell transfusions is approximately two times lower than in those with recent transfusions (Figure 6). The tracking of individual patients before and after transfusion reveals that total PK activity can increase from 2 to 10 times, potentially leading to falsely normal PK values post-transfusion. These patients are represented by matching colored dots across both plots.

We believe that the challenge of diagnosing PKD in the context of recent transfusions can be addressed by measuring the specific PK activity in isolated reticulocytes (A_R_). As previously shown (Figure 5A) A_R_ is markedly decreased in PKD. Given the extremely small number of reticulocytes received from the donor and their short-livedness in the bloodstream (1–2 days), the reticulocyte population a few days after transfusion is predominantly represented by the patient’s own reticulocytes. Thus, PK activity in these cells can be used to detect PKD even in patients with recent transfusions.

For example, in patients No. 1, 4, 16, 29, 40 (Table 1), who had received transfusions one month prior to PK activity assay, total PK activity in erythrocytes was close to the normal range, while A_R_ was markedly reduced. Given that A_R_ may also be decreased in other types of anemia (Figure 5C), it is necessary to establish a diagnostic cut-off for A_R_. At a cut-off of 52.6 IU/gHb, the sensitivity and specificity of A_R_ for differentiating PKD from other anemias was 96% (AUC of 0.99, LCL 0.976, UCL 1.008).

## 3. Discussion

In this study, we confirmed the lack of correlation between blood reticulocyte count and total PK activity in patients with genetically confirmed PKD, and for the first time provided an explanation for this observation. Specifically, we demonstrated that PK activity is already markedly reduced in reticulocytes from PKD patients compared to healthy controls (Figure 5A,B). Furthermore, the difference in PK activity between reticulocytes and mature erythrocytes is more pronounced in donors than in patients.

In normal erythrocytes the activity of certain glycolytic enzymes including PK is several times lower than in reticulocytes [12,14,26]. Consequently, it is expected that reticulocytosis may lead to falsely normal PK activity results in PKD diagnosis. However, in our cohort, PK activity remained below the normal range in PKD patients regardless of reticulocyte count (Figure 4A) except in those with recent transfusions of donor erythrocytes. To investigate this further, we isolated red blood cell fractions enriched in reticulocytes and calculated specific PK activity in both reticulocytes and mature erythrocytes. The median of PK activity in patient reticulocytes was reduced by 23 times compared to the median value for healthy controls. We also observed different A_R_/A_E_ ratios across groups: while healthy donors showed high A_R_/A_E_ values, this ratio was significantly lower in PKD patients indicating a decrease in their PK activity already at an early stage of erythroid maturation. This explains why the increase in reticulocyte count does not contribute to the overall measured PK activity in patients with PKD (Figure 4A), in contrast to healthy controls. This is in good agreement with the study of A. Zanella et al. [15]. Earlier, M. Lakomek et al. [27] reported an A_R_/A_E_ ratio of 16.2 which was the same for both patients and healthy donors. However, the lack of genetic characterization in that cohort limits direct comparison. Our findings suggest that the A_R_/A_E_ ratio may vary depending on the specific *PKLR* genotype (Table 1), providing further insight into the heterogeneity of PKD. Importantly, neither A_R_ nor A_R_/A_E_ ratio correlated with anemia severity (see Appendix A).

Interestingly, in patients with other anemias, we also observed a decrease in the specific activity of PK in both reticulocytes and erythrocytes, alongside a decreased A_R_/A_E_ ratio. However, these changes were less pronounced compared to those observed in patients with PKD (Figure 5C,D). Several hypotheses have been proposed in the literature regarding the underlying mechanisms of reduced PK activity in patients with other anemias. In hereditary spherocytosis, it has been suggested that membrane instability may cause loss of glycolytic enzyme complexes, including PK, which are localized on the erythrocyte membrane [18]. In sickle cell anemia, thalassemia, and xerocytosis, it is suggested that oxidative stress may impact the structure and/or function of the PK enzyme [22,28]. In Diamond–Blackfan anemia, it is hypothesized that cell cycle arrest in response to p53 transcription factor activation may affect glycolytic pathway enzymes, leading to a reduction in their expression and activity [24]. A key question arises regarding the threshold of residual PK activity that indicates anemia due to PK deficiency. In this study, we found that the median of residual PK activity in patients with PKD was 27%, which is consistent with the findings of S. Lenzner et al. [29], where residual activity in PKD patients ranged from 10% to 25%. According to the mathematical model of erythrocyte glycolysis proposed by M. Martinov et al. [3], the required PK activity for hemolysis is 22% and 52% for the stable and unstable protein forms, respectively. Our samples include patients with both stable and unstable forms of the mutant protein, so the median residual PK activity of 27% is generally consistent with the model calculations.

We demonstrated the sensitivity and specificity of the biochemical determining PK activity for the diagnosis of PKD using ROC analysis on a large cohort of patients with various genetically confirmed anemias and healthy donors. The method exhibited high sensitivity and specificity for PKD detection (91% and 95%, respectively). Sensitivity increased to 97% when patients with recent transfusions were excluded. Previous studies reported 90% sensitivity for PK activity measurement in patients without recent transfusions [9,17].

There is a clinical need to distinguish PKD from other anemias. However, these anemias may also have decreased PK activity (11% of patients) and PK/HK ratio (47% of patients). So we are the first to evaluate the sensitivity and specificity of PK activity and PK/HK ratio in PKD patients compared to both healthy donors and patients with other anemias. Under these conditions the PK/HK ratio has lower specificity for identifying PKD compared to PK activity (if using the lower limit of normal values for both parameters as cut-off values). The PK/HK ratio was lower for PKD patients compared to other anemias. To use this ratio effectively, a cut-off below the values for healthy donors is required. Additionally, in the case of donor erythrocyte transfusions, the PK/HK ratio may also yield false normal results, similarly to the measuring of PK activity in erythrocytes.

Many patients with PKD are transfusion-dependent, typically requiring transfusions once a month. In our cohort of PKD patients, 52% received regular donor red blood cell transfusions at intervals of ≤3 months for a long period of life. Even when PK activity is measured one month after a transfusion, the total PK activity measured can be significantly elevated (Figure 6: red, yellow, cyan dots). Bianchi P. et al. [7] proposed that the diagnosis of PKD should be considered positive when PK activity is slightly below normal in a patient who recently received a donor erythrocytes transfusion. We demonstrated that even a month after transfusion PK activity may be falsely normal and suggested an alternative solution to this issue. Our data are consistent with the study by G. Rijksen et al. [30], which approximately estimated that 4 weeks after donor erythrocyte transfusion, a patient’s blood contains only about 20% of their own cells. We suggested a method for determining PK activity during transfusions based on the specific PK activity in reticulocytes which is significantly reduced in PKD patients. One month after transfusion, the patient’s blood contains only their own reticulocytes, unlike erythrocytes. Therefore, in transfused patients, it is advisable to isolate reticulocytes and measure PK-specific activity in them.

This study has several limitations. First, it was conducted at a single center, which may limit the generalizability of the results, although this is a federal diagnostic center accepting patients from all regions of the country. Second, potential differences in analytical methods between laboratories highlight the need to establish laboratory-specific reference values. The inclusion of patients with other anemias in the control group is also important for defining more accurate differential diagnostic cut-offs. Finally, the proposed diagnostic cut-off values may require validation in multicenter studies with a large number of patients with other types of anemia before they are implemented in routine clinical practice.

## 4. Materials and Methods

### 4.1. Study Population

Whole blood samples were obtained from healthy donors and patients with hereditary molecularly confirmed anemia after obtaining written informed consent. This study was performed in accordance with the Declaration of Helsinki. All patients were followed up at the Dmitry Rogachev National Medical Research Center of Pediatric Hematology, Oncology and Immunology (Moscow, Russia). This study was approved by the Ethical Committee of that center (No. 1/2024). Medical records were reviewed to obtain information on patient characteristics, results of genetic analysis, laboratory data, and information on donor erythrocyte transfusions.

### 4.2. Reagents

The substrates and enzymes used to determine PK and HK activity were from Sigma Aldrich (Saint Louis, MO, USA), Roche (Rotkreuz, Switzerland) or TCI (Tokyo, Japan). The buffer components, phosphate-buffered saline (PBS) and inorganic salts were from Sigma Aldrich (Saint Louis, MO, USA). All compounds were ≥95% pure. Distilled water with a Milli-Q purification system (Millipore, Burlington, VT, USA) was used to prepare all solutions. Percoll for the isolation of reticulocytes in a density gradient was from Sigma Aldrich (Saint Louis, MO, USA). For reticulocyte counting, the cells were stained with a thiazole orange solution “LumiCell Reticulocyte Stain” from Lumiprobe (Moscow, Russia). Removal of leukocytes from whole blood was performed using a microcrystalline cellulose from SRL (Gurugram, India). To determine the concentration of hemoglobin, the cyanmethemoglobin method was used. β-Mercaptoethanol and EDTA for the preparation of the enzyme-stabilizing solution were prepared from Sigma Aldrich reagents.

### 4.3. Routine Hematological Studies

Routine hematological studies were performed on XN Series analyzers from Sysmex (Hyogo, Japan).

### 4.4. Genetic Analysis

Genetic analysis was performed using the Hemolytic Anemia panel or using whole-genome high-throughput DNA sequencing (NGS). The enriched DNA library was sequenced through the IIllumina NextSeq platform (San Diego, CA, USA).

### 4.5. Blood Preparation for Studies

Whole blood (5 or 10 mL) from patients and healthy controls was collected in S-Monovette vacuum tubes with K_3_EDTA from SARSTEDT (Sarstedtstraße, Germany). Removal of leukocytes from whole blood was performed using a α-cellulose column according to the method described by E. Beutler [31]. Hemolysates for PK and HK activity measurement was prepared from purified erythrocytes in an aqueous solution of β-mercaptoethanol with EDTA according to E. Beutler [31] and frozen at −80 °C until use.

### 4.6. Isolation of Reticulocytes

Separation of erythrocytes into fractions of different reticulocyte content was achieved by centrifugation in Percoll density gradient [32]. A suspension of erythrocytes purified from leukocytes was layered on 70% Percoll. After centrifugation (15 min at 1200× *g*, 30 °C), two layers of cells separated by a Percoll solution with different numbers of reticulocytes were observed in the density gradient (Figure 7). The maximum enrichment of the upper layer with reticulocytes reached 84% for patients and 19% for donors. From these two layers, 3 fractions of cells with different reticulocyte content were removed (Figure 7) and washed from Percoll by 3-fold centrifugation in PBS. The reticulocyte content in each fraction was determined using thiazole orange to stain the reticulocytes. Thiazole orange was added to erythrocyte suspensions containing reticulocytes and the samples were incubated for 30 min in the dark and then analyzed on a flow cytometer BD FACSCanto optimized for fluorescence quantitation in the visible region of the spectrum (488 nm). Hemolysates were prepared from each fraction for measuring PK activity for samples without reticulocyte isolation.

### 4.7. Measurement of Enzyme Activity in Erythrocytes

PK and HK activity in hemolysates were measured spectrophotometrically by the methods described in E. Beutler [31] using a plate photometer from Eppendorf (Hamburg, Germany) at 340 nm and 37 °C and are presented as IU/g hemoglobin (Hb). The hemoglobin concentration in the hemolysate was determined by the cyanmethemoglobin method as described in E. Beutler [31]. PK and HK activity in a suspension of erythrocytes isolated from whole blood purified from leukocytes represents the total activity of the enzymes.

### 4.8. Determination of Specific PK Activity in Erythrocytes and Reticulocytes

Specific PK activity in erythrocytes (A_E_) and reticulocytes (A_R_) was determined according to M. Lakomek et al. [12] using a graphical dependence of the measured enzyme activity in erythrocyte fractions with different reticulocyte counts versus the reticulocyte counts in these fractions. The value of activity at x = 0% and 100% (where x is the number of reticulocytes in the suspension) corresponded to the specific activity of the enzyme in erythrocytes and reticulocytes, respectively.

### 4.9. Statistical Analysis

Statistical analysis was performed using OriginPro version 2021 and MadCalc version 23.3.7 software. Diagnostic sensitivity and specificity of PK activity and PK/HK ratio were determined using ROC analysis, where values of PK activity or PK/HK ratio in patients with PKD were taken as positive results, and corresponding values of healthy donors and patients with other anemias were taken as negative results. Upper and Lower Control Limits (UCL/LCL) of 95% confidence intervals for the sensitivity/specificity of PK activity were 79.2–97.6% and 90.2–97.8%, respectively, at a cut-off of 8.4 IU/gHb, and for the sensitivity/specificity of the PK/HK ratio were 86.3–100.0% and 52.8–87.3%, respectively, at a cut-off of 10. The 95% confidence intervals for the sensitivity/specificity of A_R_ for differentiating PKD from other anemias were 78.9–99.9% and 74.0–99.0%, respectively, at a cut-off of 52.6 IU/gHb. Comparison of ROC curves was made using the DeLong method. The Kruskal–Wallis test with Dunn’s post hoc test or the Mann–Whitney test at significance level of *p* < 0.05 were used to determine statistically significant differences between the three groups and the two groups, respectively. Residual PK activity was expressed as a percentage of the median PK activity of healthy donors. To determine reference ranges for healthy donors a 95% reference interval (range between 2.5 percentile and 97.5 percentile) was calculated. To identify correlations, the Spearman correlation coefficient (r) was calculated.

## 5. Conclusions

The data from our study suggest that reticulocytosis does not significantly contribute to the observed total PK activity. This is due to the markedly reduced (23-fold) PK activity in reticulocytes of PKD patients compared to normal reticulocyte activity.A distinct difference in the ratio of specific PK activities between reticulocytes and erythrocytes is observed in PKD patients and healthy donors. Healthy donors show a more pronounced difference in PK activity between these two cell types than patients.Key factors influencing the sensitivity and specificity of PK activity include transfusions of donor erythrocytes and decreased PK activity in other types of anemia, including within reticulocytes.In patients receiving regular transfusions (every 1–3 months), PK deficiency can be diagnosed by isolating reticulocytes and measuring specific PK activity in the reticulocytes.Slightly decreased PK activity is insufficient for diagnosing PK deficiency. In 97% of cases, residual PK activity greater than 66% in patients without recent transfusions indicates anemia not related to PK deficiency.The PK/HK ratio demonstrates higher sensitivity than PK activity when diagnosing PKD relative to healthy donors. However, it is less specific for the differential diagnosis of PKD and other anemias due to potential reduced PK activity and frequently observed reticulocytosis in other anemias. For this ratio to be used effectively, it is necessary to establish a reference range that accounts for the PK/HK ratio in patients with other types of anemia.

## Figures and Tables

**Figure 1 ijms-26-08606-f001:**
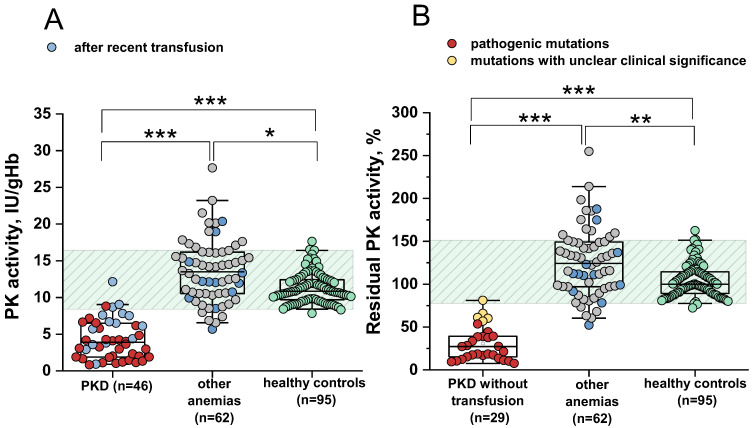
Pyruvate kinase activity in patients and healthy controls. (**A**) Pyruvate kinase (PK) activity in patients with anemias (red and gray dots represent patients with PK deficiency (PKD) and patients with other anemias, respectively, blue dots represent patients with recent transfusions) and healthy controls (green dots). (**B**) Residual PK activity in patients with PKD without recent transfusions (red and yellow dots represent patients with PKD with pathogenic mutations and mutation with unclear clinical significance, respectively), patients with other anemias (gray and blue, blue dots represent patients with recent transfusions) and healthy controls (green). Box plots: range 25–75 percentile, whisker range 2.5–97.5 percentile, median values—horizontal lines. Shaded area—reference interval (95% interval for healthy donors). *, **, ***—Differences between groups are significant at *p* < 0.05, *p* < 0.01, and *p* < 0.001, respectively (Kruskal–Wallis test, post hoc Dunn’s test).

**Figure 2 ijms-26-08606-f002:**
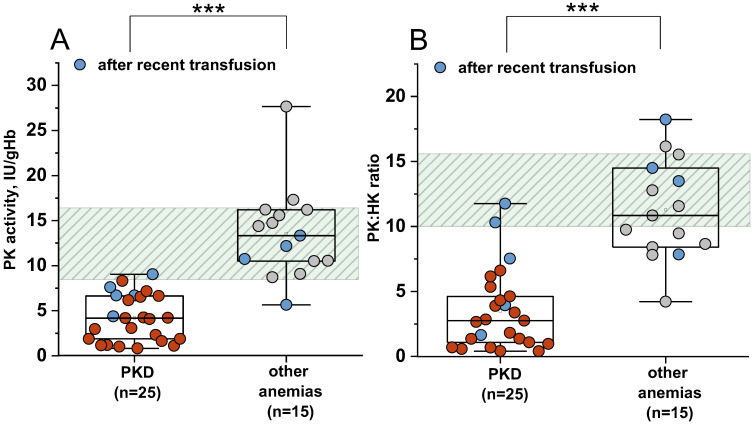
PK activity (**A**) and PK/HK ratio (**B**) in patients with PKD and other anemias. PKD—patients with pyruvate kinase (PK) deficiency (red), other anemias—patients with other anemias (grey dots). Blue dots—patients after recent transfusion; PK/HK ratio—ratio of PK activity to hexokinase (HK) activity. Box plots: range 25–75 percentile, whisker range 2.5–97.5 percentile, median values—horizontal lines. Shaded area—95% reference interval. ***—differences between groups are significant at *p* < 0.001 (Mann–Whitney test).

**Figure 3 ijms-26-08606-f003:**
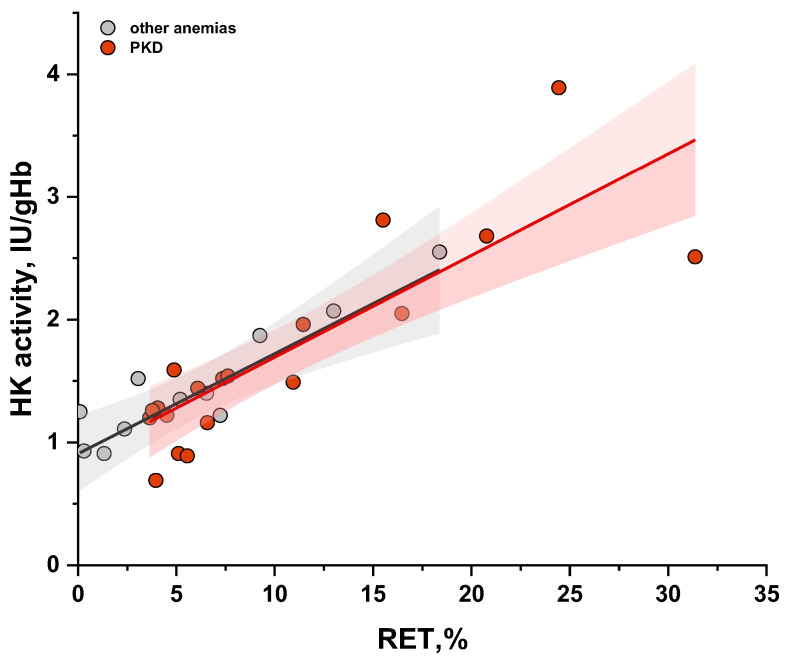
The relationship between measured hexokinase (HK) activity in erythrocytes and reticulocyte (RET) count in the blood in patients without recent transfusion. PKD—patients with pyruvate kinase deficiency (n = 20), red dots; patients with other anemias (n = 11), gray dots. The black (r = 0.79, *p* < 0.05) and red lines (r = 0.8, *p* < 0.05) represent linear fitting for other anemias and PKD, respectively, with 95% confidence interval (grey area for other anemias and red for PKD).

**Figure 4 ijms-26-08606-f004:**
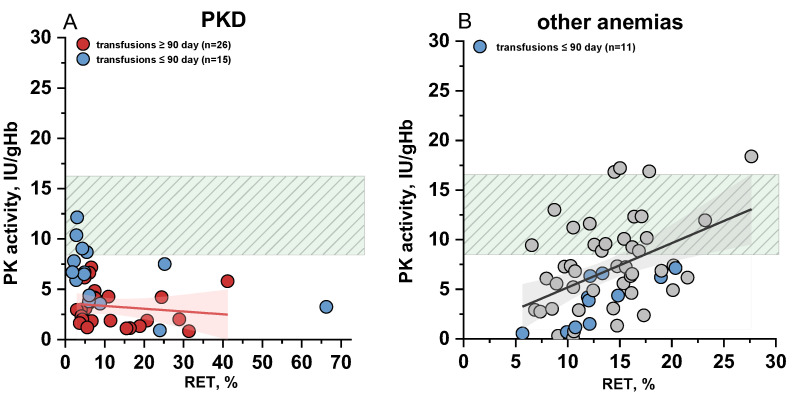
The relationship between pyruvate kinase (PK) activity in erythrocytes and reticulocyte (RET) count in the blood. (**A**) Patients with PK deficiency (PKD); (**B**) patients with other anemias. Blue dots—patients after recent red blood cell transfusions. The green shaded area on both panels represents the reference range for PK activity. The red and black lines represent linear fitting for PKD (r = −0.17 at *p* = 0.39) and other anemias (r = 0.42 at *p* = 0.0013), respectively, with 95% confidence interval (red area for PKD and grey for other anemias).

**Figure 5 ijms-26-08606-f005:**
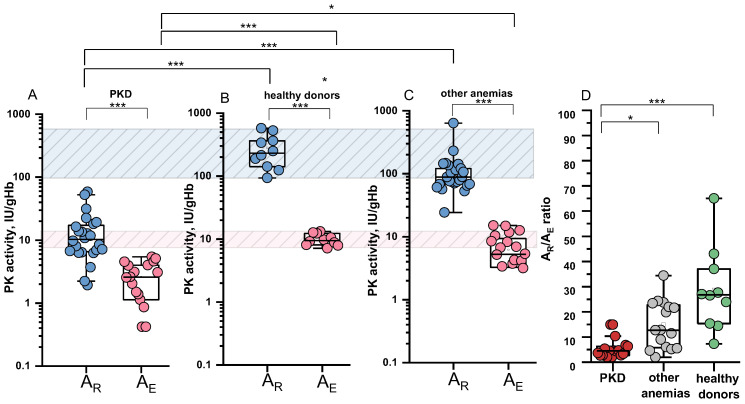
PK activity in reticulocytes (A_R_) and erythrocytes (A_E_) and A_R_/A_E_ ratio for patients and healthy donors. A_R_—blue dots, A_E_—pink dots. (**A**) Pyruvate kinase deficiency (PKD), n = 24 and n = 18 for A_R_ and A_E_, respectively; (**B**) healthy donors, n = 10 for A_R_ and A_E_; (**C**) other anemias, n = 25 and n = 20 for A_R_ and A_E_, respectively; (**D**) A_R_/A_E_ ratio for PKD (red), n = 17; other anemias (grey), n= 16; healthy donors (green), n = 10. For patients with recent transfusions, A_E_ data were excluded from the graph and calculations of the A_R_/A_E_ ratio. Box plots: range 25–75 percentile, whisker range 2.5–97.5 percentile, median values—horizontal lines. The shaded areas represent the normal reference ranges for A_R_ (blue) and A_E_ (pink). *, ***—differences between groups are significant at *p* < 0.05 and *p* < 0.001, respectively (Kruskal–Wallis test with post hoc Dunn’s test or Mann–Whitney test).

**Figure 6 ijms-26-08606-f006:**
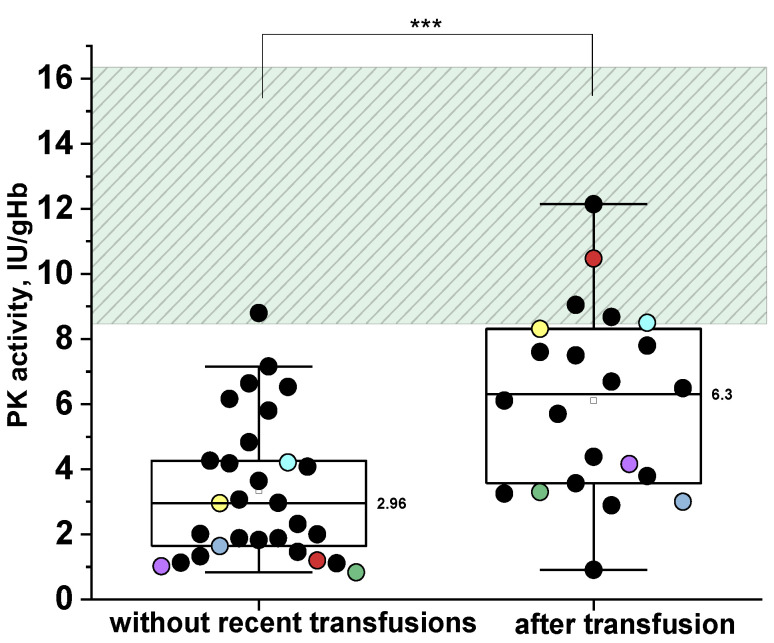
Pyruvate kinase (PK) activity in patients with PK deficiency without recent transfusions (left) (n = 29) and with recent transfusions (≤3 months prior to assay) (right) (n = 22). The same patients in the absence and presence of recent transfusion are indicated by identical colored dots. Box plots: range—25–75 percentile, whisker range—2.5–97.5 percentile, median values—horizontal lines. The shaded areas represent the normal reference range. ***—differences between groups are significant at *p* < 0.001 (Mann–Whitney test).

**Figure 7 ijms-26-08606-f007:**
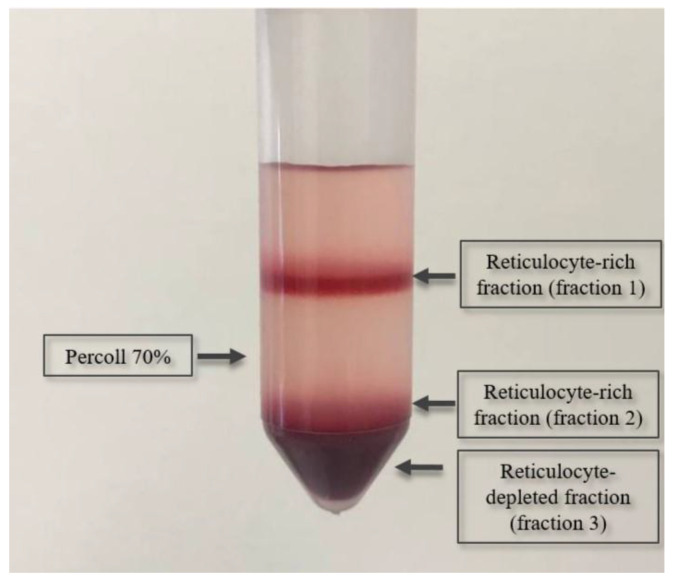
Separation of erythrocytes into fractions in a Percoll density gradient.

**Table 1 ijms-26-08606-t001:** Total and specific PK activity in erythrocytes (A_E_) and reticulocytes (A_R_) and the A_R_/A_E_ ratio in patients.

Patient	Gene Mutation	cDNA Nucleotide Substitution		Total PK Activity IU/gHb	A_R_, IU/gHb	A_E_, IU/gHb	A_R_/A_E_	PK/HK Ratio
		Allele1	Allele2					
Patients with pyruvate kinase deficiency								
Severe condition								
1. *	*PKLR*	hom c.101-1G>A		7.8	14 ± 5	7.7 ± 0.7	1.8 ± 0.7	n/d
2.	*PKLR*	hom c.401T>A		5.8 ± 0.4	n/d	n/d	n/d	n/d
3.	*PKLR*	hom c.695-2A>C		2.01 ± 0.07	n/d	n/d	n/d	n/d
4. *	*PKLR*	hom c.1079G>A		7.60	23 ± 11	7.52 ± 0.36	3.1 ± 1.5	10.3
5. *	*PKLR*	hom c.1269+1G>A		12.1	n/d	n/d	n/d	n/d
6.	*PKLR*	hom c.1529G>A		1.65	6.2 ± 0.7	1.39 ± 0.09	4.5 ± 0.6	1.37
6. *				3	n/d	n/d	n/d	n/d
7.	*PKLR*	hom c.1529G>A		1.20	2.23 ± 0.08	1.13 ± 0.01	1.98 ± 0.04	1.35
7. *				10.47	n/d	n/d	n/d	n/d
8.	*PKLR*	hom c.1529G>A		1.83	7.0 ± 0.3	1.50 ± 0.03	4.64 ± 0.11	n/d
9.	*PKLR*	hom c.1529G>A		1.13	3.7 ± 2.3	0.86 ± 0.43	4.3 ± 3.4	0.39
10.	*PKLR*	hom c.1529G>A		1.88	n/d	n/d	n/d	0.96
11.	*PKLR*	hom c.1529G>A		1.88	n/d	n/d	n/d	0.70
12. *	*PKLR*	c.101-1G>A	c.1318G>T	5.7 ± 0.2	n/d	n/d	n/d	n/d
13. *	*PKLR*	c.948C>A	c.848T>A	6.11	53 ± 7	5.08 ± 0.17	10.4 ± 1.4	7.53
14.	*PKLR*	c.1130T>C	c.1318G>T	8.8 ± 0.02	n/d	n/d	n/d	n/d
15. *	*PKLR*	c.1174G>A	c.1456C>T	7.51 ± 0.08	n/d	n/d	n/d	n/d
16. *	*PKLR*	c.1174G>A	c.1456C>T	9.05	11.7 ± 0.3	8.80 ± 0.14	1.33 ± 0.04	11.8
17. *	*PKLR*	c.1436G>A	c.487C>T	2.89 ± 0.2	n/d	n/d	n/d	n/d
18.	*PKLR*	c.1456C>T	c.1157C>T	4.18	8 ± 3	4.0 ± 0.8	2.0 ± 0.9	2.75
19. *	*PKLR*	Ex 1-2 del	c.1529G>A	3.25 ± 0.09	n/d	n/d	n/d	n/d
20. *	*PKLR*	Ex 1-2 del	c.1529G>A	3.3	12.69 ± 2.16	2.8 ± 0.25	4.5 ± 0.9	n/d
20.				0.83	n/d	n/d	n/d	0.41
21. *	*PKLR*	Ex 1-2 del	c.1529G>A	4.16	6.39 ± 2.38	4.3 ± 0.3	1.5 ± 0.7	n/d
21.				1.02	n/d	n/d	n/d	0.68
22. *	*PKLR*	c.-63G>A	c.1529G>A	6.69	n/d	n/d	n/d	8.92
23. *	*PKLR*	c.101-1G>A	c.1529G>A	3.6 ± 0.2	n/d	n/d	n/d	n/d
24. *	*PKLR*	c.460G>A	c.1529G>A	4.38	n/d	n/d	n/d	1.65
25. *	*PKLR*	c.994G>A	c.1529G>A	3.8 ± 0.4	n/d	n/d	n/d	n/d
26.	*PKLR*	c.1079G>A	c.1529G>A	0.91	1.9 ± 0.9	0.4 ± 0.3	5 ± 3	n/d
27.	*PKLR*	c.1223C>T	c.1529G>A	4	n/d	n/d	n/d	n/d
28. *	*PKLR*	c.1583A>T	c.1436G>A	6.49	n/d	n/d	n/d	n/d
29. *	*PKLR*	c.1637T>C	c.1529G>A	8.5	13.5 ± 0.12	8.27± 0.02	1.63 ± 0.015	n/d
29.				4.21	n/d	n/d	n/d	1.08
*Moderate condition*								
30.	*PKLR*	hom c.1318G>A		1.11	9.8 ± 0.9	0	n/d	0.4
31.	*PKLR*	c.932T>C	c.1456C>T	2.97	7.1 ± 0.96	2.6 ± 0.096	2.7 ± 0.4	4.31
32.	*PKLR*	c.1231G>T	c.1456C>T	4.08	19.2 ± 3.2	3.06 ± 0.25	6.3 ± 1.2	2.65
33.	*PKLR*	c.1231G>T	c.1456C>T	4.26	8.46 ± 4	3.1 ± 0.6	2.9 ± 1.4	2.84
34. *	*PKLR*	c.1130T>C	c.1456C>T	8.7 ± 0.5	n/d	n/d	n/d	n/d
35.	*PKLR*	c.1594C>T	c.1456C>T	2.96	n/d	n/d	n/d	2.35
35. *				8.31	n/d	n/d	n/d	n/d
36.	*PKLR*	c.1130T>C	c.1456C>T	6.64	16 ± 6	4.5 ± 1.2	3.6 ± 1.6	n/d
*Mild condition*								
37.	*PKLR*	c.932T>C	c.1456C>T	1.46 ± 0.17	n/d	n/d	n/d	n/d
38.	*PKLR*	c.1076G>A	c.1456C>T	4.83 ± 0.3	n/d	n/d	n/d	n/d
39.	*PKLR*	c.1181C>T	c.1456C>T	6.16	31.3 ± 1.0	4.61 ± 0.19	6.8 ± 0.4	3.88
40.	*PKLR*	c.1181C>T	c.1456C>T	7.16	18 ± 3	5.4 ± 0.6	3.3 ± 0.7	6.15
41.	*PKLR*	c.1195del	c.1456C>T	3.07	7 ± 6	2.6 ± 0.4	2.6 ± 1.0	3.38
42.	*PKLR*	c.1291G>A	c.1529G>A	6.53	59 ± 6	3.9 ± 0.4	15 ± 2	5.35
43.	*PKLR*	c.665G>A	c.1429A>G	3.64	n/d	n/d	n/d	n/d
44.	*PKLR*	c.1072G>T	c.1529G>A	1.33	6.27 ± 0.5	0.42 ± 0.2	15 ± 8	n/d
45.	*PKLR*	c.1583A>T	c.1510C>T	2.32	10.8 ± 0.4	2.03 ± 0.03	5.3 ± 0.2	1.81
46.	*PKLR*	hom c.1529G>A		2	n/d	n/d	n/d	n/d
*Median*				3.9	10.27	2.57	4.5	2.75
*Patients with other anemias*								
47.	*ALAD*	het c.375del		14.38	n/d	n/d	n/d	9.46
48.	*ANK1*	het c.5097-33G>A		19	76.1	14.8	5.15	n/d
49.	*ANK1*	het c.596dup frameshift ter		15.0	74.5 ± 1.1	0	n/d	n/d
50.	*ANK1*	het c.4104+4A>G		8.50	120 ± 20	5.2 ± 0.8	23 ± 6	n/d
	*PIEZO1*	het c.3284A>C						
51.	*ANK1*	het c.3329_3336 delinsACAAG		16.13	71 ± 9	11.7 ± 1.6	6.1 ± 1.1	n/d
52. *	*ANK1*	het c.3778T>C		14.8	110 ± 30	8 ± 2	14 ± 5	n/d
53. *	*ANK1*	het c.1814del		13.3	65 ± 17	8.7 ± 1.5	7.5 ± 1.3	14.50
54.	*ANK1*	het c.596dup frameshift ter		15.4	61 ± 4	5.3 ± 0.9	11.5 ± 2.1	n/d
55. *	*ANK1*	het c.4153C>T		11.9	136 ± 14	6.9 ± 0.8	20 ± 3	n/d
56.	*ANK1*	het c.2325dupG		13.3	115	3.36	34.4	n/d
57.	*GPI*	c.1039C>T	c.1612C>A	27.7 ± 0.5	n/d	n/d	n/d	10.8
58.	*HBB*	het c.193G>T		10.5	n/d	n/d	n/d	8.41
59. *	*HK1*	c.1951G>A	c.2128G>A	12.2	79.1 ± 1.1	9.3 ± 0.3	8.5 ± 0.3	18.22
60. *	*HK1*	homo c.34C>T		5.66	144 ± 3	5.5 ± 0.4	26.2 ± 2.0	14.2
	*HBB*	homo c.316-106 C>G						
61.	*KCNN4*	het c.940T>C		16.2	130 ± 40	8 ± 4	17 ± 10	11.6
62.	*PIEZO1*	het c.7483_7488dup		9.60	68 ± 3	3.2 ± 1.0	21 ± 7	n/d
63.	*PIEZO1*	het c.7483_7489dup		10.5	53 ± 2	4.1 ± 0.7	13 ± 2	n/d
64.	*PIGA*	c.264delA	c.715+1G>A	16.4	56.8	10.3	5.49	n/d
65. *	*RPL11*	het c.45delT		10.7	n/d	n/d	n/d	7.84
66. *	*RPS19*	het c.3G>A		9.92	80 ± 2	6.4 ± 0.2	12.5 ± 0.6	n/d
67.	*SLC4A1*	het c.1030C>T		15.6	74 ± 12	8.2 ± 1.6	9 ± 2	12.8
68.	*SLC4A1*	het c.1030C>T		10.5	88.3 ± 1.3	3.7 ± 0.3	24 ± 2	7.81
69.	*SLC4A1*	het c.749del		20.2	230 ± 50	0	n/d	n/d
70.	*SPTA1*	het c.82C>T		16.1	600 ± 400	0	n/d	n/d
71.	*SPTA1*	het c.2222A>T		11.1	150 ± 60	6.7 ± 1.9	22 ± 10	n/d
72.	*SPTA1*	het c.4019C>A		14.7	n/d	n/d	n/d	16.2
	*NF1*	het c.4614G>A						
	*KDM6A*	het c.2308C>T						
73.	*SPTA1*	het c.5645_5647del		17.3	n/d	n/d	n/d	15.5
	*FLNA*	het c.1790_1791delTT						
	*COL3A1*	het c.689A>T						
	*CFH*	het c.2407T>A						
74.	*SPTA1*	c.4339-99C>T	c.6421C>T	13.6	24.1	12.6	1.91	n/d
75.	*SPTA1*	het c.2671C>T		23.2	68.8	15.2	4.54	n/d
	*HBB*	het c.364G>C						
76.	*SPTB*	het c.566+1G>A		17.1	104 ± 3	4.3 ± 0.8	24 ± 5	n/d
77.	*SPTB*	het c.1912C>T		14.5	89 ± 13	0	n/d	n/d
	*PKLR*	het c.1456C>T						
78.	*SPTB*	het c.5800_5801insCAGG		8.72	65 ± 6	0	n/d	4.21
79.	*ANK1*	het c.4462C>T		16.2	n/d	n/d	n/d	8.65
80.	*SF3B1*	n/d		9.08	107 ± 10	8.4 ± 0.3	12.7 ± 1.3	9.75
Median				13.5	88.3	5.2	12.7	10.8
Healthy donors (n = 95)								
Median				10.85	231.5	9.4	26.7	13.5

*—Patients with recent transfusions (≤3 months prior to PK activity analysis); A_R_ and A_E_—specific pyruvate kinase (PK) activity in reticulocytes and erythrocytes, respectively; A_R_/A_E_—the ratio of reticulocyte-specific PK activity to erythrocyte-specific PK activity; PK/HK ratio—the ratio of total pyruvate kinase activity to hexokinase activity; hom—homozygous mutation; het—heterozygous mutation.

## Data Availability

The original contributions presented in this study are included in the article and Appendix A. Further inquiries can be directed to the corresponding author.

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
