# Peer review of "Pyruvate Kinase Deficiency: Markedly Decreased Reticulocyte PK Activity and Limited Specificity of the PK/HK Ratio"

_ijms, 2025, doi:10.3390/ijms26178606_

Round 1

Reviewer 1 Report

Comments and Suggestions for Authors

The paper compares the diagnostic utility of total pyruvate kinase (PK) activity and the PK:HK ratio (to hexokinase) for the identification of PK deficiency (PKD) in 46 patients with PKD and 62 with other anemias, as compared to 95 healthy donors.

My opinion is that the study fills an important practical gap in everyday hematological diagnostics: namely, in the case of reticulocytosis and/or transfusions being simultaneously present, which biochemical parameter best discriminates PKD from other hemolytic anemias. The results show that total PK, if there is the proper management of transfusions, has higher specificity than PK:HK when other anemias in the comparison group are present, with AR being highlighted as a useful marker in transfused patients.

I'll suggest the following changes to improve the quality of the manuscript.

  1. It would be helpful to rephrase the abstract to eliminate grammatical (e.g. "To avoid the influence… was proposed to measure…" with no specified subject) and stylistic flaws and to use more fluent English syntax; a straightforward "Background–Methods–Results–Conclusions" format with definite numbers (sensitivity/specificity, AUC, thresholds) would make the message clearer and more convincing.
  2. It would be helpful to fully and consistently finalize acronyms on their first appearance (AR, AE, AR/AE, PK:HK), to use a uniform decimal point (comma or period), and to visually check for possible character mixing (e.g. Cyrillic “о” in “frоzen”) that escape copy-editing.
  3. A more detailed description of the individual activities of AR and AE, which are derived from Percoll fractionations—enrichment, flow cytometric estimation by thiazole orange, reduction to IU/gHb, and PK–RET modeling—would be helpful in enabling reproducibility of the process without the need for referral to the Supplement.
  4. It would lend credibility to report the method for the ROC (e.g. DeLong for comparison of AUC), 95% confidence intervals for AUC, sensitivity/specificity and cut-offs (e.g. 8.44 IU/gHb for PK, 52.6 IU/gHb for AR), and to specify whether cut-offs were from Youden or from a pre-determined clinical value (e.g. "lower limit of normal" for PK:HK).
  5. It would be helpful to more clearly declare that the decreased specificity of PK:HK (73%) is when the comparison group contains other anemias with elevated RET and/or secondary PK diminution, along with an analytic table featuring separate entities (e.g. spherocytosis, thalassemia) and percentages of low PK:HK per diagnosis.
  6. An enhanced "Limitations" section would bring balance, mentioning single-center origin, pediatric/mixed cohort, potential differences in analytical techniques among laboratories and the requirement for external validation of cut-offs in independent cohorts.

Reviewer 2 Report

Comments and Suggestions for Authors

The article presented by the authors is devoted to solving a complex clinical problem – establishing a differential diagnosis of hemolytic anemia when pyruvate kinase (PK) deficiency is suspected. The authors propose a practical solution to this problem: measuring the specific activity of PK in reticulocytes.

The authors demonstrate that, unlike in other anemia groups, in patients with PKD (pyruvate kinase deficiency), PK activity is reduced not only in erythrocytes but also in reticulocytes. Frequent, regular transfusions of donor red blood cells in such patients can mask the true state of PK deficiency. An important practical solution is to test PK activity in reticulocytes – 2-3 days after a transfusion, the donor reticulocytes leave the circulation, meaning that the reduced PK activity in the reticulocytes will reflect values only from the patient. The use of a large clinical cohort of patients with molecularly confirmed diagnoses makes the conclusions highly reliable.

The article is wonderful illustrated – the Tables in the text and SupplM provide complete information about the patients and test results.

The article does not require any changes. 

There are a few proofreading notes:

  1. In Figure 1, the box plots have different line thicknesses (some are thin, while others are bold).
  2. In the SupplM – in the table, some data in the RET column use a comma as a decimal separator
  3. In the main text, line 218, for (r 0.42), = is missing, and the p-value is not indicated.

Round 2

Reviewer 1 Report

Comments and Suggestions for Authors

The authors made all necessary corrections by upgrading their manuscript.